# Ecological network analysis reveals cancer-dependent chaperone-client interaction structure and robustness

Geut Galai [1], Xie He [2], Barak Rotblat [1,3] & Shai Pilosof [1] ✉

Cancer cells alter the expression levels of metabolic enzymes to fuel proliferation. The mitochondrion is a central hub of metabolic reprogramming, where chaperones service hundreds of clients, forming chaperone-client interaction networks. How network structure affects its robustness to chaperone targeting is key to developing cancer-specific drug therapy. However, few studies have assessed how structure and robustness vary across different cancer tissues. Here, using ecological network analysis, we reveal a non-random, hierarchical pattern whereby the cancer type modulates the chaperones' ability to realize their potential client interactions. Despite the low similarity between the chaperone-client interaction networks, we highly accurately predict links in one cancer type based on another. Moreover, we identify groups of chaperones that interact with similar clients. Simulations of network robustness show that this group structure affects cancer-specific response to chaperone removal. Our results open the door for new hypotheses regarding the ecology and evolution of chaperone-client interaction networks and can inform cancer-specific drug development strategies.

Cancer cells reprogram their metabolism to fuel proliferation, promote an undifferentiated state, drive metastasis, overcome metabolic stress and communicate with the immune system[1]. Metabolic reprogramming is a hallmark of cancer and is executed by changes in the expression levels of metabolic enzymes. The mitochondrion is a metabolic hub playing a crucial role in metabolic reprogramming, where chaperones* (Table 1; items in the glossary are starred on first occurrence) service hundreds of clients, forming chaperone-client interaction (CCI) networks. The effect of network structure on its robustness* to chaperone targeting is key to developing cancer-specific drug therapy. However, how CCI network structure varies across different cancer tissues and how this variation affects robustness remains unknown.

The vast majority of mitochondrial proteins are synthesized in the cytosol and imported into the mitochondria in an unfolded state, after which they are folded by mitochondrial chaperones[2]. There are fifteen mitochondrial chaperones, including ATP-dependent proteases, catering to the folding and elimination of damaged and misfolded mitochondrial proteins[3]. In cases where demand for chaperone activity is higher than the available folding capacity, proteins will misfold, aggregate, and be degraded[2]. The dependency of tumor cells on particular mitochondrial chaperones is exemplified by the biological activity of small molecules targeting mitochondrial chaperones such as TRAP1[4], HSPD1[5], ClpX[6] and ClpP[7], all of which are promising anticancer compounds. Hence, predicting the consequences of targeting a mitochondrial chaperone is essential for using mitochondrial chaperone inhibitors or activators in clinical settings. However, to make predictions, we need a framework that allows quantifying variation in CCI network structure across environments and linking structure to robustness.

Ecological network* analysis is particularly suitable for analyzing interaction variation across environments[8]. In parallel, it has been argued that ecological theory can benefit cancer studies[9–11]. In this regard, studying chaperone-client interactions is analogous to

[1]Department of Life Sciences, Ben-Gurion University of the Negev, Beer-Sheva, Israel. [2]Department of Mathematics, Dartmouth College, 27 N Main St, Hanover, NH 03755, USA. [3]The National Institute for Biotechnology in the Negev, Beer Sheva 8410501, Israel. ✉e-mail: pilos@bgu.ac.il

## Table 1 | Glossary

| Term | Definition |
|------|-----------|
| Chaperone | A mitochondrial protein that is part of the agents supporting proteostasis in the mitochondria[3]. |
| Client | A mitochondrial protein interacting with a chaperone. |
| Ecological network | A mathematical graph object that represents pairwise biotic interactions (edges)—such as competition, predation, pollination—between multiple species (nodes)[54]. Networks can be depicted as matrices in which each row and column represent an entity, and matrix entries are the links. |
| Niche | The set of biotic and abiotic conditions within which an organism can perform biological functions. In the context of ecological interactions, for example, hosts are the niches of parasites. Here, we consider a chaperone's clients as its niche. |
| Niche overlap | The degree to which two species overlap in the set of conditions they can exploit. For instance, the common set of prey of two predators. Here, the set of clients that interact with two chaperones. |
| Niche separation | The processes that drive two species to minimize overlap in their niches (e.g., due to competition). Here, the tendency of two chaperones to interact with different sets of clients. |
| Nestedness | A particular network organization in which the more specialist nodes interact with proper subsets of the more generalist ones[12]. |
| Realized niche | The exploitation of a subset of all potentially exploitable resources due to some environmental or biotic constraints. Here, the realized niche of a chaperone is the subset of the clients with which it interacts. |
| Robustness | One definition of the response of a network to perturbations[27]. Can be quantified by following the proportion of nodes remaining without links following consistent node removal[29]. |
| Specialization | The extent to which a species can exploit resources out of all available ones[55]. Here, the number of clients a chaperone interacts with out of all clients. |

Due to the interdisciplinary nature of this work, we provide definitions of key terms used in this paper.

studying ecological species interaction networks. Like plant-pollinator interactions, chaperone-client interactions form bipartite networks, which describe links between two distinct sets of nodes[12]. In ecology, the environment is a strong determinant of species interactions[8,13]. Therefore, even when two species co-occur in multiple environments, they may interact in one environment and not the other[14,15]. Analogously, cancer types are distinct environments that may modulate CCI networks. Furthermore, linking structure to robustness is a long-standing theme in network ecology. For instance, in mutualistic networks, redundancy in interactions, whereby multiple insects pollinate the same plant increases network resilience because when one pollinator goes extinct, others can still pollinate the plant[16]. Hence, understanding chaperone redundancy will enable designing therapeutic protocols that will lead to the total collapse of the network or, on the contrary, to the maintenance of network integrity while affecting only a few proteins. Hence, delineating the interplay between structure and robustness in different cancer environments is crucial because controlling metabolic reprogramming via inhibitors that target specific chaperones may be cancer-specific.

Leveraging the framework of network ecology, we hypothesize that interactions between chaperones and clients are enabled by their biophysical traits alone. In this case, CCI should be similar across cancer environments, and the misfolding of different mitochondrial proteins should be tolerated similarly across cancer types. Hence, inhibiting a specific chaperone is expected to lead to the collapse (e.g., protein misfolding) of similar proteins in different tumor entities. An alternative hypothesis is that variation among cancer tissues implies variation in metabolic demands and reprogramming. In this case, the cancer environment dictates clients' dependency on a chaperone, resulting in a concomitant variation in CCIs, and, therefore, network structure across cancers. Hence, targeting a specific chaperone will lead to the collapse of different clients in different cancer types, resulting in cancer-specific cellular fates[17].

Here, we apply ecological network analysis to test CCI. We use a set of CCI networks from 12 cancer environments. The chaperones and their clients occur in all the networks, allowing us to investigate the effect of the cancer environment on interaction redundancy and structure. We specifically ask: (1) How do CCIs vary across cancers? (2) Can we use the structure of CCI in one cancer to predict interactions in another? (3) What are the consequences of structural variation to the robustness of the network in each cancer environment following the removal of chaperones? We reveal a non-random and hierarchical pattern whereby the cancer type modulates the chaperones' ability to realize their client interaction potential. Moreover, there is strong niche separation* whereby groups of chaperones interact with distinct clients. Nevertheless, there is still redundancy such that chaperones interact with similar sets of clients. Redundancy enables predicting missing CCI and increases the robustness of the networks to targeted removal of chaperones. Niche separation, redundancy, and robustness vary across cancer environments, highlighting the role of cancer type in modulating chaperone-client interactions.

## Results

### Chaperone interaction patterns are non-randomly affected by cancer type

Our networks encode interactions between 15 (Supplementary Table S2) mitochondrial chaperones and 1142 client proteins across 12 cancer environments (Supplementary Fig. S5). All chaperones and clients were present in all 12 networks. We estimated interactions based on coexpression data while normalizing sample size across cancer types (Supplementary Table S1). Comparison to protein interaction databases showed that our estimated interactions were significantly supported by experiments (see "Methods" and Supplementary Note 1).

We defined $P_c$ as the total number of clients a chaperone $c$ can interact with across cancers. We calculated the level of specialization* $S_c$ of each chaperone $c$ as $P_c$ divided by the 1142 potential protein clients, $S_c = P_c/1142$. A value of 1 indicates that the chaperone can interact with all 1142 proteins. Specialization reflects variation in chemical and physical properties that enable chaperones to interact with mitochondrial proteins independently of cancer. In ecological jargon, chaperones with low values of $S_c$ are broadly considered specialists, and those with high values as generalists. Specialization ranged from 40 to 65% ($55.5 \pm 8.1\%$) (Supplementary Fig. S1A). Because in our system, all clients were present in all cancers, the number of clients a chaperone interacts with should be equal across cancers if only biophysical properties determine interactions. Deviation from uniformity for a chaperone indicates that the cancer type affects its interactions. To test this, we calculated cancer-specific specialization—the proportion of proteins a chaperone interacts with in a given cancer environment—as $S_c^\alpha = L_c^\alpha/1142$, where $L_c^\alpha$ is the number of links of a chaperone $c$ in cancer type $\alpha$. Instead of uniform specialization, we find large

variability in chaperones' interactions across cancers (Supplementary Fig. S1B).

Following the fact that chaperones interact with different sets of clients and considering the evidence for the effect of cancer type on chaperone interactions, we calculated the number of clients a chaperone interacted with in specific cancer out of all the clients it could potentially interact with, $R_c^\alpha = L_c^\alpha / P_c$. This is analogous to the concept of realized niche* in ecology. We find a clear pattern in how cancer types affect chaperones' realized niche (Fig. 1). From the chaperone perspective: cancer environments affect the realized niche of chaperones. For example, SPG7 interacts with a high proportion of the clients that it can in thyroid cancer (THCA) but with a low proportion in breast cancer (BRCA). From the cancer perspective: chaperones vary in their realized niche within the same cancer type. For instance, in breast cancer, the CLPP chaperone realizes about 40% of its potential, while SPG7 realizes about 15%. Put together, these two observations create a weighted-nested* pattern[18,19]. That is, chaperones that interact with a small proportion of their potential clients are subsets of those that interact with a higher proportion (nestedness across rows); on the other hand, cancer environments that enable chaperones to interact with few of their potential substrates are subsets of those that enable higher proportions (nestedness across columns). The pattern of weighted-nestedness was non-random when compared to 1000 counterpart networks assembled from networks in which chaperone-clients interactions were shuffled[19] ("Methods"; Supplementary Fig. S2). Moreover, we found a similar non-random weighted nestedness in the level of specialization across cancers, $S_c^\alpha$ (Supplementary Fig. S1B).

Taken together, we find non-random structured variation in how cancer environments mediate chaperone interactions. Specifically, there is a distinct hierarchy in the specialization and realized niche of chaperones and in how cancers mediate chaperone interactions. Nestedness creates a core of chaperones that interact with most of the substrates that they can in a few cancer types (Fig. 1). We note that

unlike in ecological networks[12,13,19,20], here nestedness is not a pattern of CCIs. Instead, it is a pattern in the realization of interactions across environments.

## Protein and chaperone expression do not explain cancer-mediated interaction patterns

Nestedness could result from variation in protein expression levels across cancers. That is, when the expression of proteins is high, then chaperone expression should also be high to ensure the proper support of all their clients. This positive correlation in expression should result in more interactions and concomitantly in higher specialization and realized niche values (also see Discussion). To test this hypothesis, we compared the distributions of median protein expression values of the different cancers. The distributions were broadly similar between cancers (Supplementary Fig. S3A), implying that hierarchy in realized niche values is not the result of variation in protein expression across cancers. From the chaperones' perspective, variation in expression should result in variation in their realized niche. Although chaperones did vary in their expression levels across cancers, there were no significant correlations between expression level and realized niche values ($R_c^\alpha$) (Fig. 2, Supplementary Fig. S3B, Supplementary Table S3). Overall, our data indicate that the expression levels of chaperones and their clients cannot explain observed patterns in the realized niche and specialization. One alternative hypothesis is that there is a gradient of dependencies on the support for mitochondrial proteins between the different cancers, where some cancers are highly dependent (e.g., KIRP) and others less (LUSC).

## Chaperones interact with similar clients in different cancer environments

The non-random variation in specialization and realized interactions (Fig. 2) does not provide the complete picture because this analysis ignores the identity of the clients. To further test the effect of cancer type on chaperone interactions, we used the Jaccard similarity index to compare the identity of the clients of each chaperone $c$ between pairs of cancer types $\alpha$ and $\beta$ ($J_c^{\alpha\beta}$). We then calculated the partner-fidelity ($J_c$) of each chaperone—an ecological measure of the similarity in interaction partners of a species in different places[21]—as the median of $J_c^{\alpha\beta}$ of each chaperone $c$ (Supplementary Fig. S4A). Partner fidelity ranges between 0 (a chaperone interacts with entirely different clients in each cancer) and 1 (a chaperone interacts with the same clients across cancers). We compared each chaperone's observed partner fidelity $J_c$ with that expected at random using z-scores (sensu[21]; "Methods"). We found that although the partner fidelity of chaperones was generally low (range 0.15–0.45), it was still higher than expected at random ($z > 1.96$) for all comparisons (Fig. 3a).

These results indicate that although chaperones tend to conserve their interaction partners across cancers, they can only do so to a limited extent. Hence, client identity could influence specialization and realized niche. To test this, we correlated the median partner fidelity with the median realized niche of each chaperone (Fig. 3b). We found a positive correlation ($r = 0.64$, $P = 0.0097$), which is expected for chaperones with high realized niche values (e.g., CLPP or HSPE1) because interacting with a large proportion of the possible clients inevitably leads to high chances of interacting with the same clients. However, the trend we uncover is not trivial for chaperones with low realized niche values because the few clients a chaperone interacts with are not necessarily the same. The positive correlation indicates that chaperones that tend to interact with few of their potential clients do so with distinct clients across cancers.

## Chaperones demonstrate niche separation and redundancy in client interactions

After testing the effect of cancer on chaperone-client interactions, we now explore within-cancer effects on chaperone interactions. We

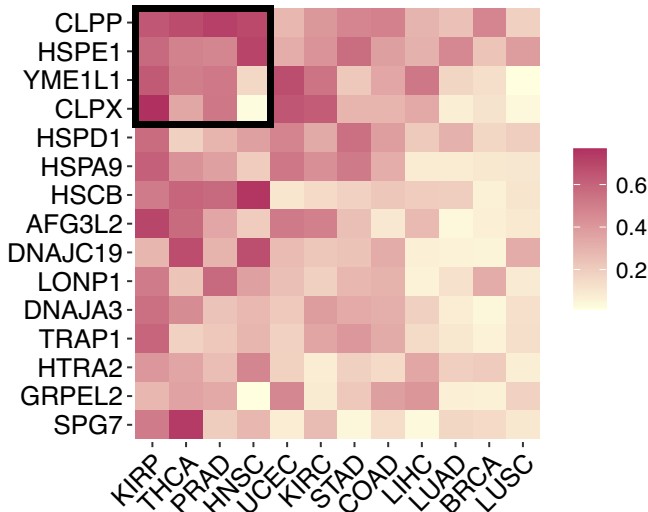

**Fig. 1 | Non-random patterns in chaperone realized niche across cancer environments.** Each square depicts the number of clients a chaperone interacts with in a specific cancer out of all the clients it could potentially interact with ($R_c^\alpha = L_c^\alpha / P_c$; realized niche, see text). The cancer environment affects the realized niche of chaperones, as is evident from the non-uniform colors in each row. Chaperones vary in their realized niche within the same cancer type, as is evident from the non-uniform colors in each column. The matrix is significantly weighted-nested, with a core of chaperones with highly realized niche values in particular environments. A core of four cancer types and four chaperones (arbitrary selection) is depicted as an example by the black square. Rows and columns are arranged by their marginal sums. A similar weighted-nested pattern was found for chaperone cancer-specific specialization (Supplementary Fig. S1B).

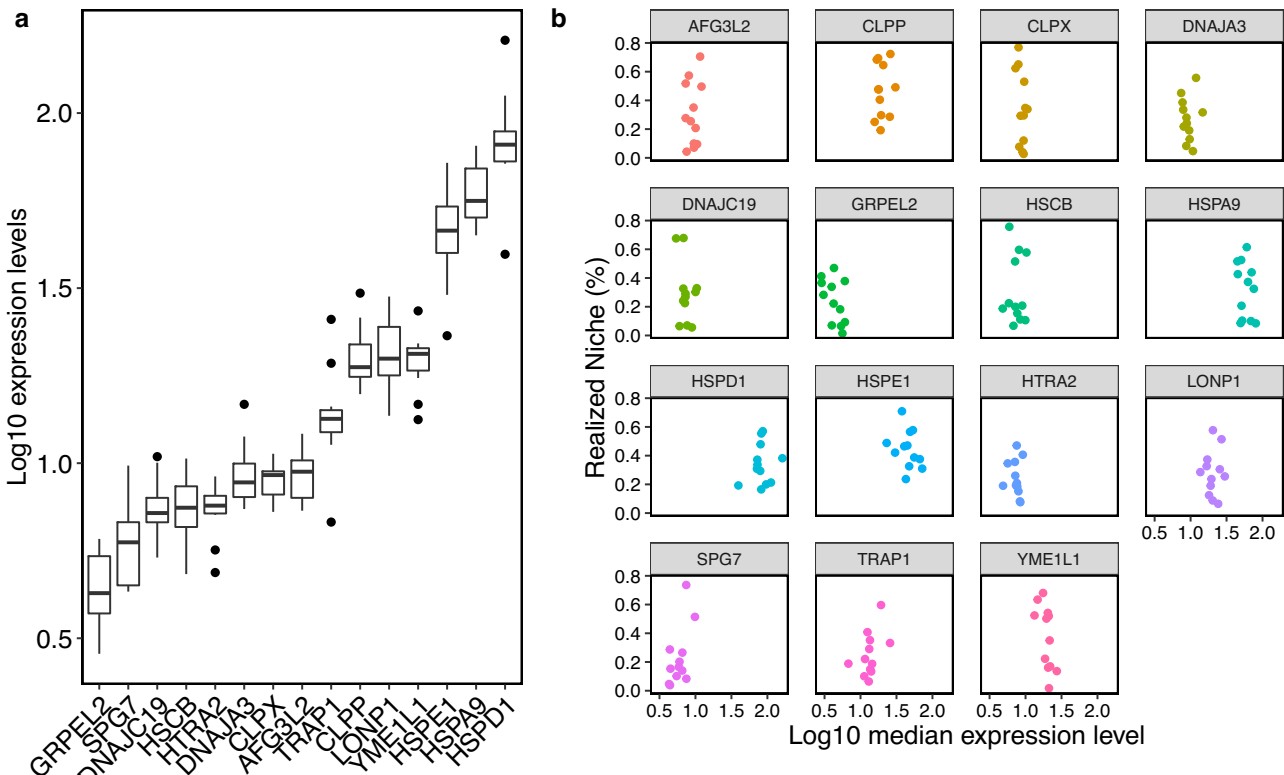

**Fig. 2 | Chaperone expression levels. a** The lof10 median values of chaperone expression levels across cancers (*n* = 12). Some chaperones are more expressed than others. Box plots: horizontal line is median, lower and upper hinges are the 25th and 75th percentiles, lower and upper whiskers are 1.5*IQR, and points are outliers **b** Each scatter plot presents the distribution of a single chaperone's

realized niche over its log10 gene expression across cancers (each data point is a cancer type). Spearman correlations combined with Bonferroni corrections between these two indices result in non-significant *p*-values for all chaperones (see Supplementary Table S3).

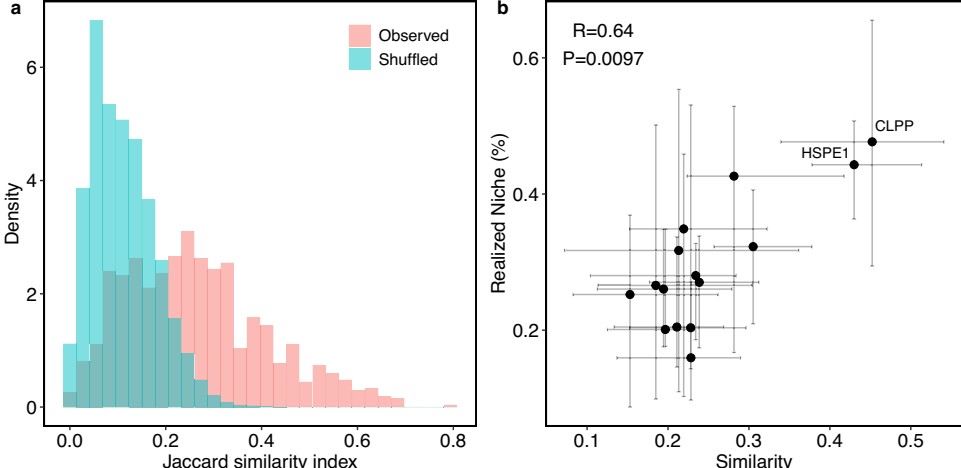

**Fig. 3 | Comparison of client identities between cancers. a** Distribution of the similarity in client identities that a chaperone interacts with between two cancer types calculated as $J_c^{\alpha\beta}$ for observed (red) and counterpart shuffled networks (blue). In the observed networks, 66 comparisons for 15 chaperones totaled 990 values of $J_c^{\alpha\beta}$. In the shuffled networks, there are 1000 values per observed value. Chaperones tend to conserve interacting proteins to a limited degree but statistically

significantly more than expected at random. **b** Each data point is a chaperone. *x*-axis values are the partner fidelity $J_c$, defined as the median of $J_c^{\alpha\beta}$ (horizontal bars are the range of $J_c^{\alpha\beta}$). *y*-axis values are the realized niche $R_c^\alpha$ (vertical bars are the range of $R_c^\alpha$). There is a positive and significant correlation (Spearman, two-tailed) between $J_c^{\alpha\beta}$ and $R_c^\alpha$.

compared the sets of clients that two chaperones *x* and *y* interact with within each cancer $\alpha$ using the Jaccard similarity index, $J_{xy}^\alpha$ ("Methods"). $J_{xy}^\alpha$ ranges from 0 (the chaperones interact with complete discordant client sets) to 1 (the chaperones interact with the same clients in the

same cancer). $J_{xy}^\alpha$ values were low and also lower than expected when compared to shuffled networks (Fig. 4a; See Supplementary Fig. S4B for within-cancer distributions). This pattern indicates ecological niche separation, whereby chaperones have little overlap in their clients.

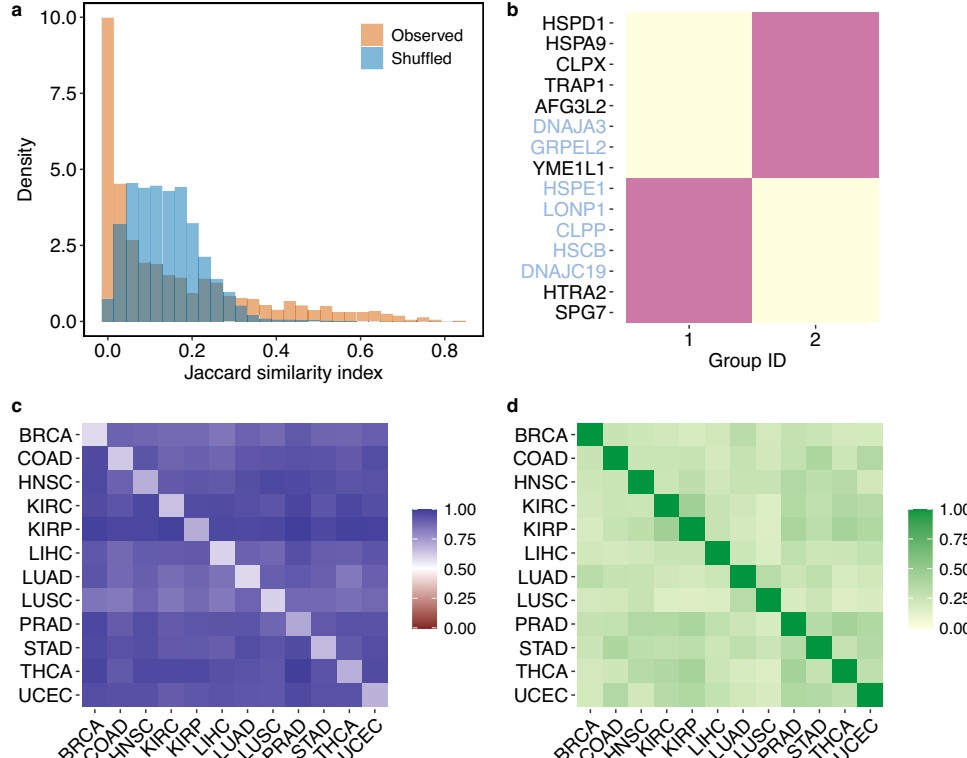

**Fig. 4 | Niche separation and redundancy in chaperone interaction partners.**
**a** The Jaccard similarity index for each pair of chaperones in a specific cancer ($J^\alpha_{xy}$; see text). The green and purple histograms depict values across all cancers for the observed and shuffled networks, respectively. The observed distribution contains 1260 pairwise comparisons (105 pairwise chaperone comparisons in 12 cancer types). The shuffled distribution contains 1000 values per observed comparison. There is a mode of lower than random $J^\alpha_{xy}$, indicating that chaperones interact with significantly different sets of clients within a cancer type. However, there is also a tail with chaperone pairs that highly overlap in the clients they interact with. **b** Using a stochastic block model, we find that chaperones are partitioned into two groups. Black and blue denominations are chaperones and co-chaperones, respectively. **c** Because of the mixed membership structure of the multilayer SBM,

we can measure to what extent information in any layer helps us predict links in the same (diagonal values) or another (off-diagonal values) layer. The y-axis and x-axis are the predicted layer and the extra layer used to help the prediction, respectively. We use 20% of the information from the predicted layer plus all of the information from another layer, which can be the same layer (diagonal) or another (off-diagonal). Cell values are the area under an ROC curve, a common measure for evaluating prediction accuracy. A value of 0.5 indicates that links can be predicted at random (50% of guessing right). We predict links with high accuracy (range 0.59–0.71 in the diagonal and 0.83–0.98 in the off-diagonal). **d** Jaccard similarity of the interactions between layers. Although layers are highly dissimilar, links are highly predictable.

Niche separation could weaken the robustness of the network to chaperone failure because if a chaperone is not functional, then its clients remain unsupported. On the other hand, robustness could be increased if some redundancy exists in client identities, such that when a chaperone is not expressed, another can take its place. Indeed, the distribution of $J^\alpha_{xy}$ also had a long tail of high values (Fig. 4a; Supplementary Fig. S4B), indicating that some chaperone pairs interact with highly overlapping sets of clients. Therefore, while the vast majority of chaperone pairs interact with distinct sets of clients, a few pairs with high redundancy may provide robustness to chaperone removal.

We investigated the interplay between niche separation and redundancy by analyzing the group structure. Chaperones within the same group interact more among themselves (redundancy) than with others (niche separation). To perform this analysis, we considered the 12 cancers as a multilayer network in which each layer is a cancer-specific CCI network. Groups are detected on all cancers simultaneously. This approach is more adequate than analyzing cancer types separately because it considers all the clients a chaperone can interact with across cancers, allowing us to detect niche separation and redundancy across all chaperones and cancer types simultaneously. Therefore, we can explicitly include the cancer type's effect on the division into groups.

We detected groups using a multilayer mixed-membership stochastic block model (SBM) with an efficient expectation-maximization

algorithm[22] (see "Methods" for details). An SBM approach has been applied to ecological networks[23–25] but not to multilayer ones. Finding community structure in multilayer networks requires finding hidden patterns in the division into groups while considering the relationships between the layers[22,26]. The chaperones were partitioned into two groups (Fig. 4b). This division is biologically meaningful because one group contains mostly co-chaperones while the other contains mostly chaperones. This result was not affected by the choice of grouping method (Supplementary Note 2).

## CCI can be predicted between cancer types despite low similarity in interactions

The SBM groups nodes based on information in multiple layers and therefore allows measuring to what extent information in any layer $\alpha$ helps us predict links in the same layer $\alpha$ or in any other layer $\beta$. To predict links within the same cancer, we remove 80% of the observed links and try to predict them using the remaining 20%. We predicted links with high accuracy (0.59–0.71; diagonal in Fig. 4c). To predict links between cancer types, we again hold out 80% of the links in cancer $\alpha$ and try to predict the missing links using the remaining 20% information from $\alpha$ plus all of the links from layer $\beta$. Here too, we predicted links with high accuracy (0.83–0.98; off-diagonal in Fig. 4c). The link prediction result is expected if cancer types have similar chaperone-client interactions. Nevertheless, cancer types have highly

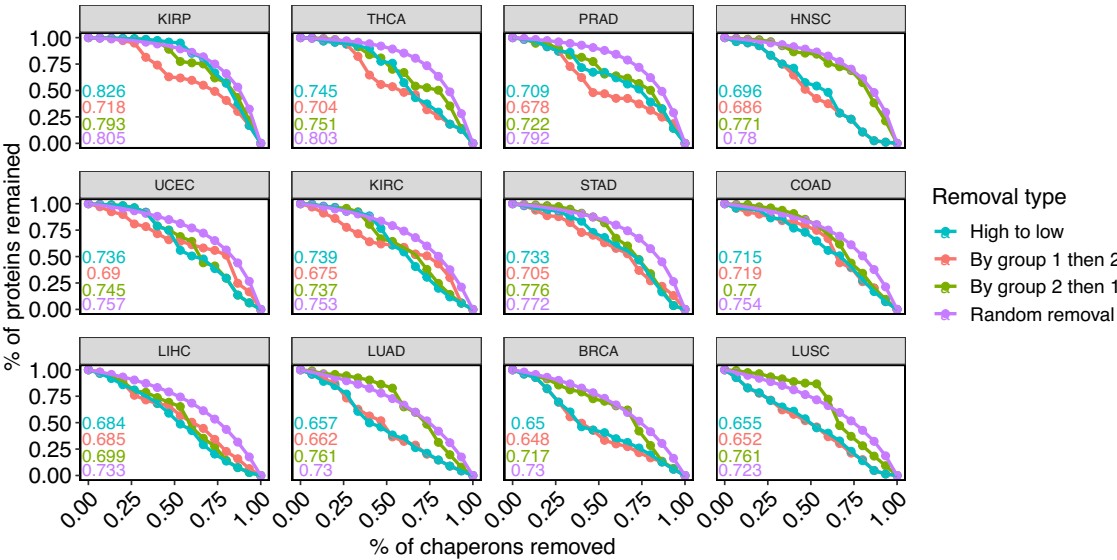

**Fig. 5 | Robustness to chaperone removal.** The collapse process of each network can be described by calculating the proportion of clients that remain connected to at least one chaperone as a function of chaperone removal. There were four scenarios of chaperone removal order. Network robustness $T$ is calculated as the area under the curve. Values of $T$ are depicted for each scenario in its corresponding color.

dissimilar interactions (Fig. 4d). Therefore, despite the strong dissimilarity in how chaperones interact with clients across cancers, there are still common patterns in the network structure that allow predicting links. These structures likely result from similar biological constraints common to all cancer types.

## Network structure affects the robustness of CCIs to chaperone removal

A primary goal of studying network structure is to understand its effect on the robustness of the networks to perturbation[27]. A relevant question in our case is how chaperone removal affects client interactions, an analysis that can provide insights into cancer therapy. To explore this question, we conducted an in-silico robustness analysis. While this analysis is common in ecology[16,28], it has not been done in the context of chaperone-client mitochondrial networks. Our algorithm simulates the targeting of particular chaperones and works as follows: In each cancer type, we removed chaperones sequentially. When a client was left without connected chaperones, it was removed and recorded as a co-extinction. We then plotted the proportion of clients remaining as a function of the proportion of chaperones removed. We calculated a robustness score, $T$, as the area under the extinction graph (Fig. 5)[29]. To explicitly link robustness to network structure, we removed chaperones in four scenarios of removal order: (1) from the most to the least connected; (2) most to least connected within module 1 and then similarly within module 2; (3) most to least connected within module 2 and then similarly within module 1; (4) randomly, to obtain a benchmark control comparison within each cancer[16]. Scenarios (2) and (3) directly link the modular structure to robustness. We expected that within each cancer, robustness scores in scenarios (1)–(3) would be lower than the random scenario and that the network would collapse most rapidly (lowest $T$) in scenario (1).

In general, the networks collapsed faster in scenarios (1)–(3) compared to random removal (Fig. 5). However, the cancer type strongly affected network collapse. While removal order had little effect on robustness in some cancer types (e.g., COAD, KIRC), in others, the effect was strong (e.g., BRCA, LUAD) (Fig. 5). Robustness is tightly linked to the general level of connectivity of the network. Cancer types that increase chaperones' realized niche value should be more robust (e.g., KIRP) than those in which chaperones realize few of their potential interactions (e.g., LUSC) (Fig. 1). We tested this hypothesis by correlating the mean realized niche values ($\bar{R}_\alpha^c$) with $T$ for each removal scenario. These correlations were statistically significant for all scenarios (Fig. 6). Therefore, the robustness of cancer networks depends on the general connectivity level of chaperones and the mesoscale group structure of chaperone-client interactions.

## Discussion

The mitochondria play a crucial role in the metabolic reprogramming of cancer cells[30]. The relationship between mitochondrial chaperones and the clients they interact with is key to cancer cell proliferation[31]. However, little was known about how mitochondrial CCI networks are structured and how this structure varies across cancer types. Portraying CCI networks as ecological multilayer networks of species interactions, we show that CCI network structure non-randomly depends on cancer type. However, despite the cross-cancer variation, knowledge of CCI in one cancer type can help us predict CCI in another. We also uncovered a group structure that provides redundancy for client support with consequences for network robustness. These results open the door for new hypotheses regarding the evolution of CCI networks and can inform cancer-specific drug therapy development.

We found a non-random pattern of weighted nestedness in the number of clients chaperones interact with across cancers. Weighted nestedness requires variation and order in that variation in both chaperones and cancer types[18]. We did not detect an effect of chaperone or protein expression levels on the chaperone's realized niche. Hence, the mechanism underlying variation in the realized niche (rows in Fig. 1) remains an open question for future research. A nested structure can arise if core chaperones are crucial for supporting essential proteins that must be sufficiently functional in all cancer types—as is the case of CLPP and HSPE1—while other chaperones are necessary only in particular cancer types to complement the function of clients supported by the core chaperones. Supporting this hypothesis is the result that chaperones with a large realized niche also tend to conserve their interaction partners across cancer types. A possible explanation for cancer-mediated variation (columns in Fig. 1) is that chaperone clients are not exclusively dependent upon them except in a particular cancer type.

We found that chaperones were separated into two distinct groups. This result is supported by a previous study in which we used a

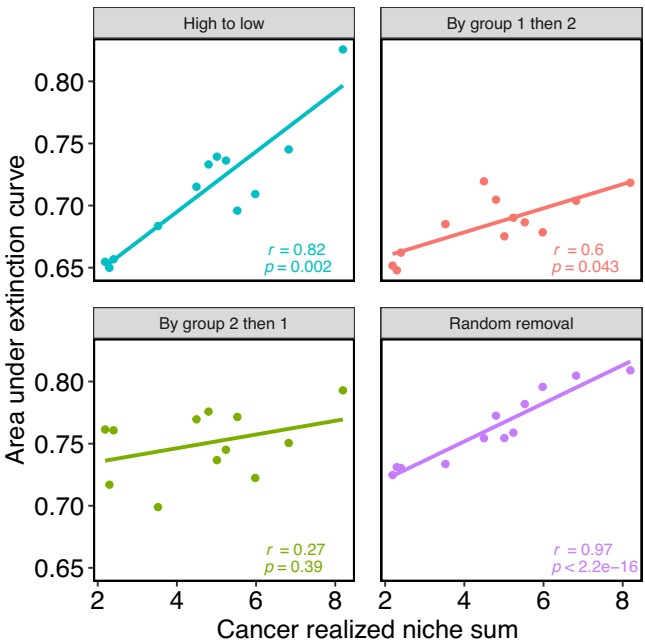

**Fig. 6 | Correlation between robustness and realized niche.** Each panel shows a scenario of the node removal order. Each data point is a cancer type. A positive and significant correlation (two-tailed Spearman correlation values within the panels) indicates that cancer environments that enable chaperones to interact with a high proportion of the clients they can potentially interact with (high realized niche), will also be more robust to chaperone removal. Random removal serves as a control: we expect a strong positive correlation because when nodes are removed randomly, robustness is a function of network size and the number of links rather than of the removed nodes.

single CCI network that ignored the cancer environment[32] and an additional analysis using unsupervised learning (Supplementary Note 2). Groups represent chaperones that interact similarly with the same clients, whom themselves interact with similar chaperones[22,23,33]. In ecology, one mechanism underlying group structure is trait matching, which emerges via coevolution[34]. In the world of proteins, grouping can emerge if there is a match in the biophysical attributes of chaperones and their clients. The fact that the groups emerge across cancers indicates that the cancer environment likely has a negligible effect on selection pressures that underlie chaperon-client coevolution. Another non-mutually exclusive hypothesis is that chaperones within the same group have complementary functions because more than one is needed to complete the folding process[35,36]. Yet another possible explanation is that selection is likely to favor redundancy because if a chaperone becomes a limiting factor, another can take its place. On the other hand, group structure also implies the specialization of a group of chaperones on a group of clients. In community ecology, this pattern of limited niche overlap* has been consistently found[23,37].

The interplay between specialization on the one hand and redundancy on the other, as manifested by group structure, has implications for network robustness. In most cancer types removing chaperones first from Group 1 resembled removal by a high-to-low degree. Moreover, in Group 1, there was a positive correlation between the chaperones' realized niche and robustness. Hence, chaperone removal will collapse more proteins. Group 1 contained a high proportion of co-chaperones, suggesting that co-chaperones contribute the most to the network's robustness. This makes them ideal candidates for drug therapy targets if the goal is to collapse the whole CCI network.

Ecological theory was used to order cancers by phenotypic axes in the "oncospace"[11]. This idea highlights the differences between cancers, supporting the cancer-dependent patterns we discover in the nestedness analysis. Nevertheless, we also showed that despite the nested pattern and the substantial dissimilarity in how chaperones interact with clients across cancers, there are common, hidden patterns in the CCI structure that allow predicting interactions in one cancer type based on another. This result can be harnessed to guide experiments for detecting specific unobserved CCI. Such predictive ability is also a tool that can help researchers understand how drugs that target CCI in one cancer may operate in another. From a basic science point of view, our findings can be used to guide studies aiming at discovering the biophysical features of proteins dependent upon a specific chaperone. In addition, these results allow for inferring possible redundancies between different chaperones.

We inferred interactions from coexpression at the mRNA level. However, genes may have evolved to be coexpressed for reasons different than a physical interaction or dependency, as is the case for the P53 target gene induced by DNA damage[38]. Additionally, while coexpression may indicate an interaction, it is important to acknowledge that a notable proportion of those interactions might be false positives or false negatives. Nevertheless, coexpression remains a powerful method for estimating interactions[39]. Moreover, we obtained empirical support that coexpression-based CCI, on the transcript level, is enriched in CCI based on protein-protein interaction data (see Supplementary note 1).

Robustness analysis is an in-silico approach that provides a first insight into stability. For instance, while inhibitors of HSP70 and HSP90 were identified and showed promising results in preclinical models, they are not routinely used in the clinic for cancer treatment as single agents[40]. RNA-based therapeutics hold the promise of being able to target any gene in the clinic[41]. In this case, deciding on the best target is key for successful treatment. Our analysis predicts which proteins will be most affected when targeting a particular mitochondrial chaperone in a particular cancer type and the potential network-wide cascading effects of targeting a chaperone. Nevertheless, our analysis does not capture the full spectrum of processes that operate in nature[42]. For instance, when a chaperone is removed, rewiring can occur such that another takes its place[43]. In addition, the importance of proteins varies across cancers. Future computational and experimental studies that consider these assumptions are needed to identify chaperones that can be used as targets for cancer therapy.

To conclude, our analysis provides a starting point for understanding variation in CCI network structure across cancer types and how structure affects network robustness. It has been argued that cancer research could benefit significantly from ecological and evolutionary theories[9,10]. Here, we apply theory and methodology from network ecology to study variation in CCI across cancer environments. By applying ecological analysis of environmental variation in ecological interaction networks and the link between structure and robustness[8,27] we provided insights into how chaperone interactions vary across cancer types, with consequences for chaperone targeting. Our study and approach can guide studies that aim to discover CCI and those that aim to test the effects of chaperone inhibitors in cancer therapy.

## Methods
### Data acquisition
Gene level transcriptome profiling (RNA-Seq) data (in the form of HTSeq-FPKM) was downloaded from The Cancer Genome Atlas (TCGA) using the Genomic Data Commons Data Portal (https://portal.gdc.cancer.gov). Of these data, we kept only the expression levels of mitochondrial proteins as listed according to the MitoCarta 2.0 database.

## Coexpression analysis and network construction

Using the raw data of each cancer tissue, we calculated a Spearman correlation between the expression levels of 15 mitochondrial chaperone genes and 1142 genes belonging to their potential protein substrates, using all available samples for that tissue. We created a chaperone-client interaction matrix (network). Chaperone-client pairs that were significantly correlated after Bonferroni correction for multiple testing received a value of 1, and non-significant correlations received a 0 (no interaction).

Cancer types varied in their sample size (Supplementary Table S1), which can create biases in the statistical power for detecting significant correlations. To ensure a fair comparison between all cancer types, we performed a bootstrap analysis to match the sample size of each cancer type to that with the least number of samples (Kidney Renal Papillary Cell Carcinoma, $n = 288$). For each cancer type, we drew 288 transcriptome samples at random without replacement (1000 attempts) and reconstructed the chaperone-client network using Spearman correlations (as described above). A correlation that was classified as significant and positive in at least 95% of the bootstrapping attempts was considered as an interaction. The results of this analysis are in Supplementary Table S1, and the networks are drawn in Supplementary Fig. S5.

## Nestedness analysis

We calculated weighted nestedness as the largest eigenvalue of a matrix, $\rho$[19]. This method is particularly suitable for comparing a matrix to its shuffled counterparts (see below for details on shuffling). Specifically, given a set of weighted interactions, the matrix with the highest weighted nestedness is that in which the distribution of matrix cell values produces the largest $\rho$[19].

## Community detection: detecting groups of chaperones

Community detection is a broad term for a suite of methods that cluster nodes to communities (also called groups, or modules)[44]. We used two different approaches to detect groups of chaperones and clients: stochastic block modeling (SBM) and modularity based on flow dynamics. These two very different methods provided almost identical node groups. We focus on SBM because this is an inferential rather than a descriptive method[45], which also enables link prediction (see below). We therefore present the methods and results of the SBM in the main text and refer the reader to Supplementary Note 2 for an extended discussion on modularity.

The general idea behind the SBM approach is that nodes are similar if they share the same kind of connection patterns to other nodes. Hence, nodes are grouped based on equivalence. In stochastic equivalence, nodes are equivalent if they connect to equivalent nodes with equal probability[44,45]. For example, two chaperones are equivalent if they both interact with equivalent clients with equal probability. There are multiple implementations of SBM for monolayer[23,46] and multilayer[22,26] networks, with some variations, but all take the same approach.

For this analysis, we used the 12 cancer types together, effectively creating a multilayer network in which each layer was a chaperone-client interaction network of a given cancer type. We detected groups using a multilayer mixed-membership SBM algorithm called multitensor[22]. This is a probabilistic method that assumes an underlying structure for $L$ layers (in our case, $L = 12$ cancers) consisting of $K$ overlapping communities. The model provides for each node a mixed membership vector of size $K$, which represents the probability that the node will be a member of each group (hence the mixed membership term). Typically in SBM-type models, nodes are assigned to a single group by choosing the group membership with the highest probability. In directed networks, each node $i$ gains two membership vectors, $u_i$ and $v_i$, which determine group membership for outgoing and incoming links, respectively (for undirected networks, $u = v$). To

consider the bipartite nature of the layers, we used a directed network with links going from the chaperones to the clients. Using a directed network creates a distinction between the different sets of nodes (chaperones and clients), effectively setting the probability of links within the same node set in $u$ and $v$ to 0. As noted in the original paper, using directed networks bears a close mathematical relationship to models that generate bipartite weighted graphs[22].

For each layer $\alpha$, a $K \times K$ affinity matrix $w^{(\alpha)}$ describes the density of edges between each group. The expected number of edges in layer $\alpha$ from $i$ to $j$ is then given by the bilinear form:

$$M_{ij}^{(\alpha)} = \sum_{k,l=1}^{K} u_{ik} v_{jl} w_{kl}^{(\alpha)}. \tag{1}$$

Multitensor and other SBM models[26] require the number of groups to be predefined. We set the range of the number of groups to $K \in (2, 15)$ (there are 15 chaperones). Because SBM models are probabilistic, the output includes a maximum-likelihood estimate, which we can use to choose the optimal number of communities. For each $K$, multitensor uses an expectation-maximization algorithm (EM) to find the maximum likelihood of the group assignment. However, maximum-likelihood estimates will always increase with an increase in the number of parameters. Therefore, we use the Bayesian Information Criterion (BIC) to penalize for the increase of parameters and avoid overfitting communities. We chose the $K$ with the lowest BIC score, which gives us $K = 2$[47].

## Link prediction

The stochastic formulation of SBMs forms generative models, which can be used for inference[44,45]. The multitensor[22] and similar models developed for multilayer networks[26] group nodes based on information contained in multiple layers. They therefore provide a mathematically principled way to define the interdependence between layers and measure to what extent information on one layer helps us predict links in another.

In all link prediction analyses, we set the number of communities to 2 since this was the best grouping according to the BIC score. To predict links within the same cancer, we calculate $M_{ij}^{\alpha}$ in a focal layer $\alpha$ to predict the existence of an edge between $i$ and $j$. We hold out 80% of the observed edges and use the other 20% of the information to do the prediction. To predict links in one cancer using information from another, we also hold out 80% of the links from a focal layer $\alpha$, and then use the remaining 20% from $\alpha$, plus all of the links from another layer $\beta$ to predict the retained links.

At the current state-of-the-art[22,26], it is impossible to predict missing links in $\alpha$ using solely information from $\beta$ because multilayer SBM models use matrix factorization, which requires partial information from the target layer. However, by holding out 80% from $\alpha$, we ensure that the analysis is insightful because leaving only 20% of links makes prediction challenging.

For both within- and between-layer predictions, we repeated the process 20 times, with 5-fold cross-validation. We evaluated the quality of link prediction using the area under the ROC curves[48] and averaged results across the 20 runs. The diagonal AUC values in Fig. 4c are for the within-layer prediction, and the off-diagonal ones are for the layer-pair prediction.

## Network shuffling

We wanted to ensure that the network properties we find for empirical networks (e.g., nestedness, similarity in interactions, Jaccard indices) are not a result of random processes but rather are biologically significant. Hence, we compared the results of empirical networks to those obtained from analyses of 1000 counterpart shuffled networks. This is a common procedure in the study of ecological networks[49–51]. We shuffled the interaction values of each CCI matrix using the

'curveball' algorithm[52], which constrains the number of client associations (for chaperones) and chaperone associations (for proteins) to that observed in the empirical networks. This algorithm is highly conservative, making it difficult to detect statistically significant results; or, in other words, it increases the likelihood of type II error[53]. This increases our confidence that the statistically significant results we find are true biological processes.

Using the shuffled networks, we determined the statistical significance of weighted nestedness using a one-tailed test, as is common in ecological networks[19,50]. We compared the empirical $\rho$ to its shuffled counterparts using the following formula:

$$p - value = \frac{count(\rho_{shuffled} > \rho_{observed})}{N_{simulations}} \quad (2)$$

To calculate the statistical significance of measures at the node level (i.e., the Jaccard index calculations), we used z-scores as follows[21].

$$z_i = \frac{J_i^{empirical} - avg(J_i^{shuffled})}{SD(J_i^{shuffled})}, \quad (3)$$

where $avg(J_i^{shuffled})$ and $SD(J_i^{shuffled})$ are the mean and standard deviation of the Jaccard index obtained from the shuffled networks. Hence, a positive (negative) z-score suggests that similarity is higher (lower) than expected from random CCIs. The significance of each empirical value was determined at the 0.05 level: a z-score >1.96 or < −1.96 indicates that the index is greater or lower than the random expectation, respectively.

### Software development
The software in this paper was implemented using R programming language (version 4.1) and Python (version 2.7) with Linux environment. Network analysis was done using the R packages 'bipartite' (version 2.15) and 'vegan' (version 2.5-7).

### Reporting summary
Further information on research design is available in the Nature Portfolio Reporting Summary linked to this article.

## Data availability
Gene level transcriptome profiling (RNA-Seq) data (in the form of HTSeq-FPKM) was downloaded from The Cancer Genome Atlas (TCGA) using the Genomic Data Commons Data Portal (https://portal.gdc.cancer.gov). Human protein expression data was downloaded from the string-db.org data base (v11.5) and from published papers. These data and the processed network data are available in figshare (https://doi.org/10.6084/m9.figshare.22779755.v2). The data underlying the figures and tables are provided in the Source Data file and on figshare. All code and data are also available on the GitHub repository: Galai, G., He, X., Rotblat, B. & Pilosof, S. Ecological network analysis reveals cancer-dependent chaperone-client interaction structure and robustness in the mitochondria, authors, title (this paper), https://github.com/Ecological-Complexity-Lab/cancer-networks, https://doi.org/10.5281/zenodo.8245737, 2023. Source data are provided with this paper.

## Code availability
All code and its description are available in the following GitHub repository: https://github.com/Ecological-Complexity-Lab/cancer-networks.

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

## Acknowledgements

We thank Prof. Peter Mucha for valuable suggestions on the stochastic block models analysis. We thank Dr. Liron Levin for help with data preparation. This work was supported by research grants from the ISF (Israel Science Foundation): 1281/20 to S.P. and 1436/19 to B.R. B.R. also acknowledges support from The Israel Cancer Association. X.H. was supported by the joint NIH-NSF-NIFA Ecology and Evolution of Infectious Disease award R01-TW011493.

## Author contributions

S.P. and B.R. conceptualized the study. G.G. curated the data. G.G. and X.H. analyzed the data. S.P. acquired funding and supervised the study. G.G. and S.P. developed figure visualizations. All authors discussed and interpreted the results. G.G., B.R. and S.P. wrote the original draft. All authors contributed to the final writing and editing.

## Competing interests

The authors declare no competing interests.
