## [Peer Review File · Nature Communications]

REVIEWER COMMENTS

Reviewer #1 (Remarks to the Author):

The authors investigated how chaperon-client interactions (CCIs) vary across cancers. They identified networks (CCINs) characterized by niche separation, redundancy, and robustness, which vary across cancer environments, highlighting the role of cancer type in modulating chaperone-client interactions. Simulations of network robustness show that this modular structure strongly affects cancer-specific response to chaperon removal. Their results open the door for new hypotheses regarding the ecology and evolution of CCINs and can inform cancer-specific drug development strategies.

The results herein presented are novel and add precious contribution to this field. The paper is very well written and easy to read, the concepts, even if tough to comprehend for researchers that do not belong to this specific field, are presented in a practical way. However, some minor issues need to be addressed before this manuscript could be considered for publication in Nature Communications.

The manuscript is centered on theoretical discussion of the computational data. It seems like perspectives on practical applications of the findings are lacking. The authors only outlined that these findings may be used in cancer-specific drug development, but they did not focus on the possible specific applications, nor they discussed the currently used pharmacological strategies acting on chaperones to treat cancer. They should better introduce this topic in the introduction section and deepen it in the discussion section.

Moreover, it would be useful for an easier interpretation of the results if the authors could add some graphical illustration of the CCINs they found, showing nodes and edges, maybe inserting them in supplementary materials.

The authors should doublecheck the text since some typos are present throughout it (e.g., can[c]er-specific, some capital letter is missing after a sentence's full stop).

In conclusion, I think that this manuscript needs a minor revision before publication in Nature Communications.

Reviewer #2 (Remarks to the Author):

In this manuscript, Galai et al build chaperone-client networks for different cancer types and then perform an 'ecological' analysis of these networks. I think that the work is original and well written. However, I have several methodological concerns that ultimately put into question the

relevance/robustness/generality of the results and conclusions of the manuscript. For this reason I cannot recommend this manuscript for publication in Nature Communications.

My first concern is about the network building process. All of the results in the manuscript hinge on the assumption that proteins that are co-expressed interact. As the authors acknowledge this is a weak point in their manuscript and I agree. The problem is that I cannot get past this bold assumption to understand whether many of the results the authors find are a by-product of their assumption which they do not demonstrate in any place to be correct. I agree that co-expression is necessary for proteins to interact. However, (and for Nature Communications) we need more reassurance that this data is correct. I am not an expert on this data, but, for instance for HSP70 there is at least a study (PLoS Biol. 2020 Jul 20;18(7):e3000606. doi: 10.1371/journal.pbio.3000606. eCollection 2020 Jul.) that maps clients of this chaperone (which is not included unfortunately in the present study). My conclusions from this work (I am obviously not an expert) is that overall chaperones have a well-defined set of clients, and while these interactions might not all happen within the same cancer type, the actual interactions (measured through co-expression) should be a subset of the total possible clients. This picture does not seem to be consistent with the results found in the manuscript in terms of nestedness, since in those cancer types in which 'promiscuous' chaperones have a smaller set of interacting clients, this should be a large overlapping set with respect of those cancer types in which co-occurrences are more common. Without a clear validation of the network is it impossible for me to say if the findings are real or an artifact of the way the networks have been constructed (see below a paragraph on the validation).

As the authors say, despite these issues, their 'pipeline' could be adopted as a method for future studies. This is true, but if this is the case, the results are not reliable and therefore I think this is a manuscript for a more specialized journal highlighting the types of conclusions you "could" draw from this type of analysis.

For the results (and conclusions) to be more reliable the authors should come up with a way to estimate the false positive and false negative rates for their co-occurrence analysis even for a few chaperones for which some clients are known. Without this test, I am afraid that this remains a possible approach that still needs to be validated.

A second major concern for me is the analysis on the group structure of the networks. The authors use modularity maximization to obtain groups of chaperones and clients. I agree that this was a 'top notch approach' 10-15 years ago but not anymore. I agree that being modular is a structural property of the network. However, modularity maximization is a 'descriptive, heuristic' approach to understanding large network structure, but as network science studies in the last 10 year plus show, if we want to draw conclusions from the 'set of groups' that best describe the network structure we observe, you should use an inferential approach (I really encourage the authors to take a look at Tiago P. Peixoto, "Descriptive vs. inferential community detection: pitfalls, myths and half-truths", arXiv: 2112.00183). Indeed, the development of group-based models (that generalize the concept of dense groups with few inter-group connections to other types of patterns of connections and define groups as nodes having the same role) to infer structural properties of networks has evolved a lot in the last 10 years and there are plenty of tools available to obtain groups that explain network structure in a superior way to that of

modularity maximization methods which have a large number of caveats, including the resolution limit which is very likely affecting the dataset the authors have (with very few chaperones). For multi-layer bipartite networks there have been different approaches proposed (de Bacco et al. <https://journals.aps.org/pre/abstract/10.1103/PhysRevE.95.042317>, <https://www.nature.com/articles/s41467-022-34714-7>) or (Tarrés-Deulofeu et al. <https://journals.aps.org/pre/abstract/10.1103/PhysRevE.99.032307>, and I believe an application to an ecological dataset <https://academic.oup.com/pnasnexus/article/1/3/pgac055/6590842>).

Again, from a network science point of view, the robustness analyses could be affected by the use of modularity maximization approaches that should be avoided.

Finally, I fully understand that this is probably a disappointing report for the authors. However, I hope they understand that this is nothing personal. I think that the work has potential, but right now it is just not compelling.

Reviewer #1

The authors investigated how chaperon-client interactions (CCIs) vary across cancers. They identified networks (CCINs) characterized by niche separation, redundancy, and robustness, which vary across cancer environments, highlighting the role of cancer type in modulating chaperone-client interactions. Simulations of network robustness show that this modular structure strongly affects cancer-specific response to chaperon removal. Their results open the door for new hypotheses regarding the ecology and evolution of CCINs and can inform cancer-specific drug development strategies.

The results herein presented are novel and add precious contribution to this field. The paper is very well written and easy to read, the concepts, even if tough to comprehend for researchers that do not belong to this specific field, are presented in a practical way. However, some minor issues need to be addressed before this manuscript could be considered for publication in Nature Communications.

Thank you for this positive feedback. We have addressed the minor issues as described below.

The manuscript is centered on theoretical discussion of the computational data. It seems like perspectives on practical applications of the findings are lacking. The authors only outlined that these findings may be used in cancer-specific drug development, but they did not focus on the possible specific applications, nor they discussed the currently used pharmacological strategies acting on chaperones to treat cancer. They should better introduce this topic in the introduction section and deepen it in the discussion section.

Thank you for this comment. As suggested, we added to the introduction a paragraph with this explanation (lines 26-34). Following our new analysis on prediction interactions, we also added to the discussion an explanation on how the computational approach can be applied (e.g., lines 270-277 and 284-294).

Moreover, it would be useful for an easier interpretation of the results if the authors could add some graphical illustration of the CCINs they found, showing nodes and edges, maybe inserting them in supplementary materials.

We have now added the figures of the networks to the SI and we mention that figure in the Results and Methods section. (Fig S5).

The authors should doublecheck the text since some typos are present throughout it (e.g., can[c]er-specific, some capital letter is missing after a sentence's full stop).

Thank you for noticing the errors. We have proof-read the entire manuscript.

In conclusion, I think that this manuscript needs a minor revision before publication in Nature Communications.

Reviewer #2

In this manuscript, Galai et al build chaperone-client networks for different cancer types and then perform an 'ecological' analysis of these networks. I think that the work is original and well written. However, I have several methodological concerns that ultimately put into question the relevance/robustness/generalizability of the results and conclusions of the manuscript. For this reason I cannot recommend this manuscript for publication in Nature Communications.

Thank you for this constructive review and the positive feedback on originality and writing. We took into careful consideration the methodological concerns raised and addressed them in this revision. Doing so has improved the basis of the work and its results tremendously. We are more confident in the way we obtained the networks. The new analysis reinforced our previous one and even allowed us to add link prediction, providing a new angle to the work. We hope these compelling results are now convincing enough of the validity and strength of this work.

My first concern is about the network building process. All of the results in the manuscript hinge on the assumption that proteins that are co-expressed interact. As the authors acknowledge this is a weak point in their manuscript and I agree. The problem is that I cannot get past this bold assumption to understand whether many of the results the authors find are a by-product of their assumption which they do not demonstrate in any place to be correct. I agree that co-expression is necessary for proteins to interact. However, (and for Nature Communications) we need more reassurance that this data is correct. I am not an expert on this data, but, for instance for HSP70 there is at least a study (PLoS Biol. 2020 Jul 20;18(7):e3000606. doi: 10.1371/journal.pbio.3000606. eCollection 2020 Jul.) that maps clients of this chaperone (which is not included unfortunately in the present study). My conclusions from this work (I am obviously not an expert) is that overall chaperones have a well-defined set of clients, and while these interactions might not all happen within the same cancer type, the actual interactions (measured through co-expression) should be a subset of the total possible clients. This picture does not seem to be consistent with the results found in the manuscript in terms of nestedness, since in those cancer types in which 'promiscuous' chaperones have a smaller set of interacting clients, this should be a large overlapping set with respect of those cancer types in which co-occurrences are more common. Without a clear validation of the network is it impossible for me to say if the findings are real or an artifact of the way the networks have been constructed (see below a paragraph on the validation).

As the authors say, despite these issues, their 'pipeline' could be adopted as a method for future studies. This is true, but if this is the case, the results are not reliable and therefore I think this is a manuscript for a more specialized journal highlighting the types of conclusions you "could" draw from this type of analysis.

For the results (and conclusions) to be more reliable the authors should come up with a way to estimate the false positive and false negative rates for their co-occurrence analysis even for a few chaperones for which some clients are known. Without this test, I am afraid that this remains a possible approach that still needs to be validated.

Thank you for this comment. In essence, the concern raised is if co-expression, as reflected in mRNA levels of two genes, indicates protein-protein interactions. A body of work shows this is indeed the case (Wyrick and Young 2002, van Dam et al. 2018, Paci et al. 2021, Shemesh et al. 2021), and specifically, that co-expression predicts protein-complex stoichiometry in the OXPHOS system (van Waveren and Moraes 2008). The parallel of this concern in ecology is that co-occurrence does not necessarily mean an interaction. However, when it comes to observing interactions, molecular cell biology is fundamentally different from ecology because at the molecular scale, observing interactions requires experiments and even then, not in all types of experiments a physical interaction can be observed. **Therefore, at the current state-of-the-art, coexpression data is used to infer interactions, as is mentioned in the studies we cited and others.**

Nevertheless, we took a step further and quantified the extent to which our coexpression-inferred links are supported by experimental data. First, we selected the three most studied chaperones (HSPD1, CLPP and TRAP1) and obtained experimental data for them from specific papers. We found that 12-29% of their coexpression-inferred interactions were validated experimentally. **Considering that these experiments were preformed using cell lines growing in culture and based on co-immunoprecipitation in lysates (HSPD1), using chemical activator (CLPP) or inhibitor (TRAP1), which are different than chaperone-client interactions occurring in human tumors, this result is highly satisfying.** Nevertheless, to show that this experimental support is more significant than expected, we randomly sampled from the mitochondrial proteins a set of interactions whose size was equal to the number of interactions each chaperone had. We repeated this 1,000 times and recorded how many interactions had experimental evidence. We found that the empirical evidence for our co-expression-inferred interactions is statistically significantly greater than for random links within the mitochondria.

Because there is a bias in experimental studies towards certain proteins (as there is in ecology for certain species), we further quantified the extent to which all of our coexpression-inferred links are supported by experimental data. We obtained experimental interaction data from the STRING database and found that 1-31% of a chaperone's interaction were also observed in experiments. To evaluate if this support is "high", we compared it to the support that we would expect for interaction with a random set of proteins from the human genome. **This test showed unequivocally that the proportion of validated interactions is greater than the support we would expect for links with random proteins.**

These results are summarized in the Supplementary Information due to limited space in the Methods section, but can move it to the Methods if necessary.

A second major concern for me is the analysis on the group structure of the networks. The authors use modularity maximization to obtain groups of chaperones and clients. I agree that this was a 'top notch approach' 10-15 years ago but not anymore. I agree that being modular is a structural property of the network. However, modularity maximization is a 'descriptive, heuristic' approach to understanding large network structure, but as network science studies in the last 10 year plus show, if we want to draw conclusions from the 'set of groups' that best describe the network structure we observe, you should use an inferential approach (I

really encourage the authors to take a look at Tiago P. Peixoto, “Descriptive vs. inferential community detection: pitfalls, myths and half-truths”, arXiv: 2112.00183). Indeed, the development of group-based models (that generalize the concept of dense groups with few inter-group connections to other types of patterns of connections and define groups as nodes having the same role) to infer structural properties of networks has evolved a lot in the last 10 years and there are plenty of tools available to obtain groups that explain network structure in a superior way to that of modularity maximization methods which have a large number of caveats, including the resolution limit which is very likely affecting the dataset the authors have (with very few chaperones). For multi-layer bipartite networks there have been different approaches proposed (de Bacco et al.

<https://journals.aps.org/pre/abstract/10.1103/PhysRevE.95.042317>,

<https://www.nature.com/articles/s41467-022-34714-7>) or (Tarrés-Deulofeu et al.

<https://journals.aps.org/pre/abstract/10.1103/PhysRevE.99.032307>, and I believe an application to an ecological dataset

<https://academic.oup.com/pnasnexus/article/1/3/pgac055/6590842>).

Again, from a network science point of view, the robustness analyses could be affected by the use of modularity maximization approaches that should be avoided.

This is an important comment, which led to a new series of analyses, guided by the provided references. We agree that there is a fundamental conceptual difference between the two approaches to community detection. Because our goal was originally to describe structure, we used Infomap. Although Infomap is a state-of-the-art method for community detection, it is still a descriptive heuristic, as argued by Tiago Peixoto (arXiv: 2112.00183). Because SBM is not an expertise of our lab, we consulted with Prof. Peter Mucha (Department of Mathematics, Dartmouth College). Together with his PhD student Xei He, we performed the analysis. We added Xei as a co-author on the paper.

To shift from a descriptive analysis to an inferential one we used an implementation of one of the methods suggested by the reviewer (de Bacco et al. 2017 PRE). We provide a full description of the analysis in the methods. Briefly, this method is a multilayer mixed-membership stochastic block model with an efficient expectation maximization algorithm that can perform two tasks: 1) detect the community structures for the 15 chaperones of interest, and 2) predict links for the 12 different cancer types that we want to study. The latter task is what really differentiates between a descriptive and an inferential approach.

Our previous analysis of community detection with Infomap showed that chaperones are divided into two major communities plus another community that contained a single chaperone (SPG7). **The SBM grouping split the chaperones into two groups, which were virtually identical to the ones identified by Infomap.** The only difference was that with Infomap, SPG7 was clustered on its own while in the SBM it was clustered with another group. **The fact that two fundamentally different approaches resulted in the same community structure reinforces the analysis and points to a strong signal in the data.** Therefore, instead of throwing out the Infomap analysis, we kept it in the Supplementary Information.

Per the reviewer’s comments we wanted to take full advantage of the *inferential* abilities of the SBM. Hence, we used the SBM to perform link prediction within and

between layers (cancer types). The prediction ability (measured with AUC) ranged from [0.59-0.71] when predicting links within the same cancer and [0.83-0.98] when predicting links based on information in another cancer. This result is surprising because cancer layers differed substantially in their links (Fig. 4D). Therefore, despite strong dissimilarity in the way chaperones interact with clients across cancers, there are likely common patterns that allow predicting links. This result can be harnessed to guide experiments for detecting specific unobserved CCI. Such predictive ability is also a tool that can help researchers understand how drugs that target CCI or metabolic pathways in one cancer may operate in another. From a more basic science point of view, this analysis can be used to answer questions such as what are the biophysical features of proteins dependent upon a specific chaperone. In addition, these results allow inferring possible redundancies between different chaperones.

Finally, we redid the robustness analysis. Because the chaperone groups have not changed, the results were maintained.

Overall, these analyses, performed using the methods suggested by the reviewer, reinforced our previous results and provided a new angle to the study.

Finally, I fully understand that this is probably a disappointing report for the authors. However, I hope they understand that this is nothing personal. I think that the work has potential, but right now it is just not compelling.

We appreciate the honesty and we are certain that this is not personal. The review is not at all disappointing; if anything, it has made the work much stronger and we thank you for that.

References

- van Dam, S., U. Vösa, A. van der Graaf, L. Franke, and J. P. de Magalhães. 2018. Gene co-expression analysis for functional classification and gene-disease predictions. *Briefings in bioinformatics* 19:575–592.
- Paci, P., G. Fiscon, F. Conte, R.-S. Wang, L. Farina, and J. Loscalzo. 2021. Gene co-expression in the interactome: moving from correlation toward causation via an integrated approach to disease module discovery. *NPJ systems biology and applications* 7:3.
- Shemesh, N., J. Jubran, S. Dror, E. Simonovsky, O. Basha, C. Argov, I. Hekselman, M. Abu-Qarn, E. Vinogradov, O. Mauer, T. Tiago, S. Carra, A. Ben-Zvi, and E. Yeger-Lotem. 2021. The landscape of molecular chaperones across human tissues reveals a layered architecture of core and variable chaperones. *Nature communications* 12:2180.
- van Waveren, C., and C. T. Moraes. 2008. Transcriptional co-expression and co-regulation of genes coding for components of the oxidative phosphorylation system. *BMC genomics* 9:18.
- Wyrick, J. J., and R. A. Young. 2002. Deciphering gene expression regulatory networks. *Current*

opinion in genetics & development 12:130–136.

REVIEWERS' COMMENTS

Reviewer #1 (Remarks to the Author):

In the revised manuscript, the Authors satisfactorily addressed my questions related to the potential applications of their work, as well as the request to add some graphical illustration to better explain the results. The Authors also checked other minor concerns, and now I feel that this work is worthy of being published.

Reviewer #2 (Remarks to the Author):

I really thank the authors for the great effort they have put into addressing my comments in detail. They have convinced me that their results are compelling and valuable to the community.

The network group analysis is now strong and consistent. Hopefully others will learn from their detailed analysis.

With respect to the co-expression interaction analysis, I understand that while the 'signal' is there, that is co-expression often indicates interaction, it is also true that they have a non-negligible fraction of false positive and false negatives. The authors should say this explicitly in their article, so that future work about robustness of the results taking into account possible errors in the data could be made to further strengthen their methodology.

In any case, I think that this is very nice work that should be published in this journal.